# Knockdown of RRM1 with Adenoviral shRNA Vectors to Inhibit Tumor Cell Viability and Increase Chemotherapeutic Sensitivity to Gemcitabine in Bladder Cancer Cells

**DOI:** 10.3390/ijms22084102

**Published:** 2021-04-15

**Authors:** Xia Zhang, Rikiya Taoka, Dage Liu, Yuki Matsuoka, Yoichiro Tohi, Yoshiyuki Kakehi, Mikio Sugimoto

**Affiliations:** 1Department of Urology, Faculty of Medicine, Kagawa University, 1750-1 Ikenobe, Miki-cho, Kita-gun, Kagawa 761-0793, Japan; zhangxia@med.kagawa-u.ac.jp (X.Z.); y-matsuoka@med.kagawa-u.ac.jp (Y.M.); ytohi@med.kagawa-u.ac.jp (Y.T.); president-yk@kagawa-u.ac.jp (Y.K.); micsug@med.kagawa-u.ac.jp (M.S.); 2Department of General Thoracic Surgery, Faculty of Medicine, Kagawa University, 1750-1 Ikenobe, Miki-cho, Kita-gun, Kagawa 761-0793, Japan; dgliu@med.kagawa-u.ac.jp

**Keywords:** RRM1, shRNA, adenoviral vector, gemcitabine, gene therapy, chemotherapeutic sensitivity

## Abstract

RRM1—an important DNA replication/repair enzyme—is the primary molecular gemcitabine (GEM) target. High RRM1-expression associates with gemcitabine-resistance in various cancers and RRM1 inhibition may provide novel cancer treatment approaches. Our study elucidates how RRM1 inhibition affects cancer cell proliferation and influences gemcitabine-resistant bladder cancer cells. Of nine bladder cancer cell lines investigated, two RRM1 highly expressed cells, 253J and RT112, were selected for further experimentation. An RRM1-targeting shRNA was cloned into adenoviral vector, Ad-shRRM1. Gene and protein expression were investigated using real-time PCR and western blotting. Cell proliferation rate and chemotherapeutic sensitivity to GEM were assessed by MTT assay. A human tumor xenograft model was prepared by implanting RRM1 highly expressed tumors, derived from RT112 cells, in nude mice. Infection with Ad-shRRM1 effectively downregulated RRM1 expression, significantly inhibiting cell growth in both RRM1 highly expressed tumor cells. In vivo, Ad-shRRM1 treatment had pronounced antitumor effects against RRM1 highly expressed tumor xenografts (*p* < 0.05). Moreover, combination of Ad-shRRM1 and GEM inhibited cell proliferation in both cell lines significantly more than either treatment individually. Cancer gene therapy using anti-RRM1 shRNA has pronounced antitumor effects against RRM1 highly expressed tumors, and RRM1 inhibition specifically increases bladder cancer cell GEM-sensitivity. Ad-shRRM1/GEM combination therapy may offer new treatment options for patients with GEM-resistant bladder tumors.

## 1. Introduction

Bladder cancer is the most common malignancy of the urinary tract and the 10th leading cause of malignancy worldwide [1]. Up to 75% of bladder cancers appear as frequently recurring non-muscular invasive tumors, and 15–30% of them progress to muscle-invasive cancers [2]. Radical cystectomy (RC) is the mainstay therapy for patients with muscle-invasive cancers [3]. Despite well-performed surgeries, cure rates with surgery alone are only between 52.0% and 72.9%, for organ-confined disease [4,5]. With progression to metastatic bladder cancer, systemic chemotherapy is considered the standard treatment. Recently, gemcitabine (GEM)—an effective cytotoxic drug—was approved for the treatment of various solid tumors [6]. In addition, combination chemotherapy with GEM and cisplatin (CDDP), exhibiting higher activity and fewer adverse effects, is a first-line treatment for patients with metastatic bladder cancer [7]. However, while the initial response to combination chemotherapy is high, long-term progression-free and overall survival rates are still low [8]. To improve these unsatisfactory results, new treatment drugs and strategies have gained much attention in recent years. 

Ribonucleotide reductase M1 (RRM1)—the large subunit of ribonucleotide reductase (RNR)—plays a central role in the process of DNA synthesis [9]. At the same time, RRM1 is the molecular target of GEM [10]. GEM is an analogue of deoxycytidine, which is phosphorylated by deoxycytidine kinase and further by a nucleoside monophosphate kinase, generating difluorodeoxycytidine 5′-diphosphate. This phosphorylated product binds to the binding site and inactivates the RRM1 subunit, whereas the triphosphorylated form is incorporated into DNA chain and inhibits DNA synthesis [11]. Studies have shown that GEM-resistance in various cancers is associated with an increased expression of RRM1 [12,13]. Several clinical studies have confirmed the relationship between high expression of RRM1 and adverse clinical outcomes in patients with advanced bladder cancer [14,15]. Of such patients who were treated with first-line GEM plus platinum combination chemotherapy, those with low expression of RRM1 showed a higher treatment response rate [16], whilst those with high RRM1 expression had a significantly greater risk of disease progression and death [15]. Therefore, inhibiting RRM1 may become a new treatment strategy in patients with bladder cancer, not only to inhibit cell viability but also to reduce GEM-resistance. 

Previously, we demonstrated that an adenoviral vector expressing short hairpin RNA that targets RRM1 (Ad-shRRM1) had antiproliferative activity against non-small cell lung cancer cells [17]. In the present study, we further explored the potential of Ad-shRRM1, this time in bladder cancer cells and found that Ad-shRRM1 effectively downregulated RRM1 expression, resulting in suppression of bladder cancer cell proliferation in vitro and in vivo. Moreover, combined treatment with Ad-shRRM1 and GEM showed effective antitumor activity in bladder cancer cells.

## 2. Results

### 2.1. RRM1 Gene Expression Is Higher in Bladder Cancer Cell Lines

The normalized RRM1 gene expression ratio was evaluated in nine human bladder cancer, six lung cancer, and six malignant mesothelioma cells (Figure 1A). RRM1 gene expression varied from 28.6% (T24) to 153.5% (J82) (median: 116.9%) in bladder cancer cells. Compared to the lung cancer or malignant mesothelioma cell lines, bladder cancer cells showed significantly higher RRM1 gene expression (96.7 ± 8.1% vs. 47.2 ± 10.2%, *p* < 0.005, Figure 1B).

### 2.2. GEM Resistance in Bladder Cancer Cell Lines 

To investigate whether bladder cancer cells are resistant to GEM, we assessed the GEM sensitivity of six bladder cancer cells (Figure 2). Among these cell lines, there was a tendency for greater resistance to GEM, with higher RRM1 gene expression. 

Two high-RRM1 gene expression cells, 253J and RT112, showing relatively higher resistance to GEM, were selected for subsequent experiments.

### 2.3. Ad-shRRM1 Effectively Downregulates RRM1 Expression in Bladder Cancer Cells

Two RRM1-expressing cells, 253J and RT112, were treated with Ad-shRRM1 at a multiplicity of infection (MOI) of 10. Ad-shRRM1 effectively knocked down RRM1 mRNA and protein expression in both RRM1 highly expressed cancer cell lines, in a time-dependent manner (Figure 3).

### 2.4. Ad-shRRM1 Inhibits Cell Viability in RRM1 Highly Expressed Bladder Cancer Cells

The inhibitory effect of Ad-shRRM1 on RRM1 highly expressed cancer cells was evaluated. The percentage of viable cells significantly decreased in both the RRM1 highly expressed cancer cells infected with Ad-shRRM1, three days after infection. Infection with Ad-shRRM1 strongly reduced the percentage of viable cells in RT112 and 253J cell lines, in a time-dependent manner (Figure 4).

### 2.5. Ad-shRRM1 Inhibits Tumor Growth in Bladder Cancer Cells Xenografts

The effects of Ad-shRRM1 on bladder tumor growth were evaluated in vivo using RT112 xenografts in nude mice. The tumor volumes at 31 days were 901 ± 476 mm^3^ in the Ad-shScramble-treated group and 197 ± 65 mm^3^ in the Ad-shRRM1-treated group (Figure 5A). The tumor volume in the Ad-shRRM1-treated group decreased significantly from 10 days after injection, compared to the tumor volume in the Ad-shScramble-treated group (Figure 5B). These results further confirmed the antitumor effect of Ad-shRRM1 in vivo.

### 2.6. Ad-shRRM1 Specifically Increases GEM Sensitivity in Bladder Cancer Cells 

Even though Ad-shRRM1 in itself showed potent inhibitory effects on cell proliferation, combination treatment with Ad-shRRM1 and GEM reduced the 50% inhibitory concentration (IC_50_) of GEM to a greater extent, in both of the RRM1 highly expressed bladder cancer cells. When GEM was administered after Ad-shRRM1 infection at a MOI of 20, these cells’ sensitivity to GEM showed significant increase, namely 25.5 times in 253J and 2.6 times in RT112 (Figure 6A,B). Conversely, no such changes in cell sensitivity to cisplatin (CDDP) were observed in either of the cell lines (Figure 6C,D). These results indicate that Ad-shRRM1 specifically enhances GEM sensitivity in RRM1 highly expressed bladder cancer cells.

## 3. Discussion

The RRM1 gene encodes the large and catalytic subunit of RNR, an enzyme essential for the conversion of ribonucleotides into deoxyribonucleotides [19,20]. Deoxyribonucleoside diphosphates are further phosphorylated into deoxyribonucleoside triphosphates for de novo DNA synthesis and DNA repair processes. RRM1 is overexpressed in various types of cancers, including bladder cancer, and is associated with cell growth, migration, tumor development, and metastasis [21]. It is a highly regulated enzyme consisting of two homodimer subunits, RRM1 and RRM2 [20]. Reid et al. verified that RRM1 silencing was more effective in inhibiting cell growth than RRM2 silencing [22]. In agreement with Reid et al., we showed that the effective inhibition of RRM1 expression with Ad-shRRM1 significantly reduced cell viability in vitro (Figure 4) along with in vivo growth of tumors that overexpress RRM1 in bladder cells (Figure 5A,B). Ad-shRNA knockdown strongly inhibited the viability of RRM1-expressing bladder cancer cells, 253J and RT112. The mechanism for such inhibition was proven by other members of our faculty, to be the inhibition of cell proliferation and induction of apoptosis in lung cancer cells [17], which was further confirmed in a recent study of multiple myeloma cells [23]. Using siRNA as an inhibitor of RRM1, Sagawa et al. showed that inhibition of RRM1 triggered significant growth inhibition and apoptosis in multiple myeloma cells. These results provide evidence that RRM1 plays an important role in cell proliferation, tumorigenesis, and tumor growth 

In addition to inhibiting cell viability, we proved in this study that Ad-shRRM1 can increase cell sensitivity to GEM. Ad-shRRM1 inhibited the 253J and RT112 cell proliferation in a dose-dependent manner. The cell viability was 86.9 ± 1.2%, 77.5 ± 3.5% and 68.2 ± 1.6% for the 253J infected with Ad-shRRM1 at 5, 10, and 20 MOI 4 day after infection (Figure 6A). The cell viability was 87.7 ± 1.5%, 74.6 ± 4.1% and 58.0 ± 0.7% for RT112 cell infected with Ad-shRRM1 at 5, 10, and 20 MOI 4 day after infection (Figure 6B). Double increase in Ad-shRRM1 exposure resulted about 9% and 12% cell inhibition for 253J and RT112 cell respectively. When in combination with GEM treatment, in case of Ad-shRRM1 exposure at MOI of 20, the sensitivity to GEM (IC50) increased significantly and was 27.5 times higher in 253J cell and 7712 times higher in RT112 cell (Figure 6A,B). On the other hand, no such changes in IC_50_ were observed in the CDDP treatment groups (Figure 6C,D). Combination therapy with Ad-shRRM1 and GEM specifically induces a synergistic cytotoxicity in RRM1-expressing bladder cancer cells and could achieve the same effect with a reduced dosage of either one of the individual drugs. This would be beneficial in clinical application, as it could result in reduced side-effects experienced by patients. 

These preclinical results may provide a novel treatment option for patients with bladder cancer, using either Ad-shRRM1 alone or in combination with GEM. GEM has been used to treat a variety of solid tumors, including pancreatic, breast, ovarian, and small cell lung cancers [24,25,26] for over 20 years. In addition, the intravesical instillation of GEM has been tested in two animal models and in two phase 1 trials for patients with non-muscle invasive bladder cancer. As a results, it was concluded that local and systemic side-effects were minimal [27]. Furthermore, combination chemotherapy with GEM and CDDP (GC) has become an effective first-line treatment for metastatic bladder cancer patients [28] and is widely used in bladder chemotherapy [29]. However, majority of patients treated with GC develop drug resistance within a few years, limiting its efficacy [30]. In this context, resistance to GEM remains the major challenge, causing adverse clinical outcomes in patients with bladder cancer [14,15,16]. 

In the present study, the RRM1 gene was shown to be a good target for inhibition of expression and Ad-shRRM1 has the potential to become very useful in clinical application. However, some limitations would need to be overcome first. Highly expressed RRM1 is not the only reason for GEM resistance. Factors such as multiple membrane transporters, target enzymes, enzymes involved in the metabolism of gemcitabine, and alterations in the apoptotic pathways may confer the sensitivity/resistance to this drug [11]. Although a high level of immune reactivity to viral infection has limited the use of adenovirus vectors in patients, it is the first effective in vivo gene delivery vector and has been widely used to express transgenes [31,32]. Recently, adeno-associated virus (AAV) has been shown to be beneficial in overcoming immunogenicity [32]. Despite AAV having a smaller payload capacity, it is sufficient for carrying the shRNA against RRM1. Moreover, due to the unique efficacy of Ad-shRRM1 in GEM-resistant bladder cancer cells, direct administration of Ad-shRRM1 will make it easier to perform gene therapy in bladder cancer patients.

## 4. Materials and Methods

### 4.1. Cell Lines 

Nine human bladder cancer cells (253J, EJ-1, HT1197, J82, Ku7, RT112, RT4, T24, and TCCSUP), 7 lung cancer (A549, EBC, H358, H69, LUDLU, MAC10 and RERF-AI), and 5 malignant mesothelioma cells (H2052, H28, MESO1, MESO4, and MSTO) were obtained from ATCC^®^ through the Japanese official distributor and investigated for gene expression. All cancer cells were cultured in RPMI 1640 medium supplemented with 10% fetal bovine serum and 1% penicillin-streptomycin, and the cells were incubated at 37 °C in a 5% CO_2_ atmosphere.

### 4.2. Real-Time RT-PCR for mRNA Expression

mRNA expression was determined using quantitative real-time PCR (qPCR). Briefly, total RNA from cells was isolated using TRIzol RNA isolation reagent (Life Technologies, Carlsbad, CA, USA). First-strand cDNA synthesis was performed using the TaqMan Reverse Transcriptase Kit (Applied Biosystems, Branchburg, NJ, USA). Real-time quantitative PCR was performed using the StepOnePlus Real-Time PCR System (Applied Biosystems, Foster City, CA, USA). Primers and probes were purchased from the Assays-on-Demand Gene Expression Assay (RRM1 assay ID: Hs00168784_m1; GAPDH assay ID: 4326317E, Applied Biosystems). The comparative threshold cycle method was used to calculate gene expression in the sample, relative to the value in the cells using GAPDH as an internal control for normalization of gene expression among samples. Each assay was performed in triplicate.

### 4.3. Western Blot Analysis for Protein Expression

Cells were harvested and prepared in Cell Lysis Buffer M (Wako Pure Chemical Industries, Ltd., Osaka, Japan), according to the manufacturer’s instructions. Rabbit RRM1 antibody (10526-1-AP; Cosmobio, Tokyo, Japan 1:500) was used as the primary antibody. A monoclonal antibody against GAPDH (AC74, Sigma-Aldrich, Tokyo, Japan, 1:2500) was used to control sample loading. The membranes were then incubated with horseradish peroxidase (HRP)-labeled secondary antibodies for 1 h. Signals were developed using the ECL chemiluminescence kit (Amersham Biosciences, Buckinghamshire, UK).

### 4.4. Construction of Adenoviral Vectors

A replication-deficient recombinant adenoviral vector expressing shRNA, targeting RRM1 (forward strand: 5′-RRM1-siRNA1 sense strand [GGAUAUGGUCCUAUAGGUU] + loop [UAGUGCUCCUGGUUG] + RRM1-siRNA1 antisense strand [AACCUAUAGGACCAUAUCC] + polymerase III terminator [UUUUUU]) under the control of the human U6 promoter (Ad-shRRM1), was constructed using the COS-TPC method, as previously described [17,33]. Ad-shScramble, an adenoviral vector expressing shRNA against the scrambled sequence (5′-UCUUAAUCGCGUAUAAGGCTT-3′) was also constructed to serve as a negative control. Constructed adenoviral vectors were amplified in HEK293 cells and purified by CsCl ultracentrifugation [17,33].

### 4.5. Cells Viability Assay

The in vitro cell viability test was performed with a 3-(4,5-demerthylthiazol-2-yl)-2, 5-diphenyltetrazolium bromide (MTT) assay, as previously described [33]. Briefly,cells were seeded at a density of 4000 cells/well in 96-well culture plates. After 24 h, the medium was removed and the cells were infected at MOI of 10 or 20, for 1 h. Cell viability was determined by MTT assay using a Cell Proliferation Kit I (Roche, Mannheim, Germany) at different time points. The cell viability in each well was measured in terms of optical density at a wavelength of 570 nm, with 750 nm as the reference wavelength. Each cell viability assay was performed in triplicate.

### 4.6. Cell Sensitivity to GEM 

Regarding cell sensitivity to GEM, we performed an MTT assay to investigate the in vitro drug concentration that inhibited cell growth by 50% (IC_50_). Cancer cell suspensions were seeded in 96-well culture plates at a concentration of 4000 cells/well and incubated at 37 °C. To roughly select the cells resistant to GEM, cells were treated with GEM at different concentrations and cell viability evaluated with the MTT assay, after 72 h. IC_50_ values were calculated using the formula described by Beck et al. [18]. Two cell lines exhibiting relatively higher resistance, 253J and RT112, were selected for the subsequent study (Figure 2).

To investigate the influence of Ad-shRRM1 on GEM-sensitivity, 253J and RT112 cells were seeded in 96-well plates at a concentration of 4000 cells/well. Twenty-four hours later, cells were infected with control virus or Ad-shRRM1 at MOI of 10 or 20, respectively. The next day, cells were treated with GEM at concentrations ranging from 0 to 100 μmol/L. After 72 h, the cells were exposed to the test agents and evaluated using the MTT assay.

### 4.7. Xenograft Tumor Model in Nude Mice

The in vivo experiments were performed as previously described [33]. Briefly, tumor xenografts were prepared by subcutaneously implanting tumors derived from RT112 cells into the backs of 6-week-old male nude mice (BALB/cA Jcl-nu/nu, CLEA, Japan). When the tumor volume reached approximately 100 mm^3^, the mice were randomly divided into two groups (five mice/group): Ad-Scramble and Ad-shRRM1. Intratumoral injection with adenoviral vectors (at 2 × 10^9^ PFU) was performed every 4 days. Tumor growth was monitored every 4 days for 30 days by measuring tumor size with a caliper. The tumor volume was calculated using the following formula: tumor volume = (length) × (width)^2^ × 0.5. Animal experiments were performed in accordance with the Guide for the Care and Use of Laboratory Animals from Kagawa University (No: 19637-1).

### 4.8. Statistical Analysis

Data are presented as the mean ± standard deviation. The independent Student’s *t*-test or analysis of variance was used to compare continuous variables between the groups. Statistical significance was set at *p* < 0.05.

## 5. Conclusions

We elucidated the potential of Ad-shRRM1 as a therapeutic agent for bladder cancer and demonstrated that Ad-shRRM1 inhibits bladder cancer cell viability and enhances GEM-sensitivity, by reducing RRM1 expression. Although further extensive studies are required, combination therapy with Ad-shRRM1 and GEM could be considered as a new effective and safe treatment for patients with GEM-resistant bladder cancer.

## Figures and Tables

**Figure 1 ijms-22-04102-f001:**
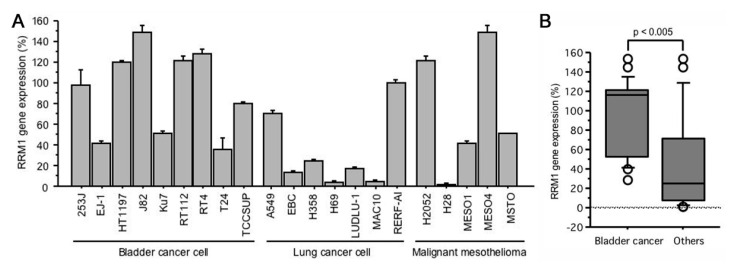
(**A**) RRM1 gene expressions in human bladder cancer, lung cancer, and malignant mesothelioma cells; (**B**) RRM1 gene expressions in bladder cancer cells is significantly higher than that of other cells. RRM1 gene expression quantification was performed using GAPDH as an internal control for normalization and expressed as a value in reference to the RRM1 expression of 253J (100%).

**Figure 2 ijms-22-04102-f002:**
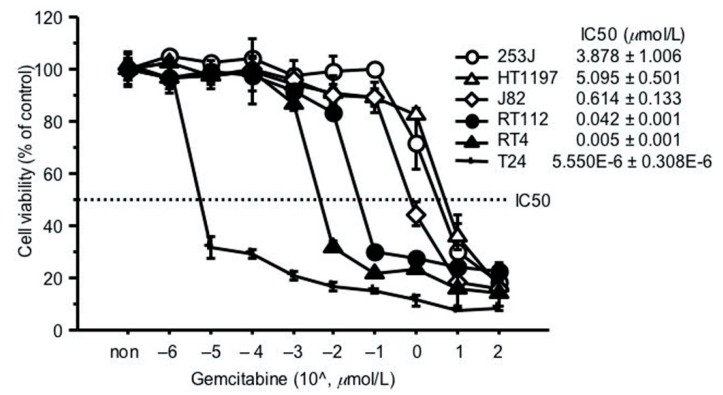
GEM resistance in bladder cancer cell lines. Sensitivity to GEM was represented as the in vitro drug concentration that inhibited cell growth by 50% (IC_50_). 4000 cancer cells were seeded in a 96-well culture plate 24 h before assay. Cells were treated with GEM at different concentrations and cell viability evaluated by MTT assay, 72 h later. IC_50_ values were calculated using the formula described by Beck et al. [18].

**Figure 3 ijms-22-04102-f003:**
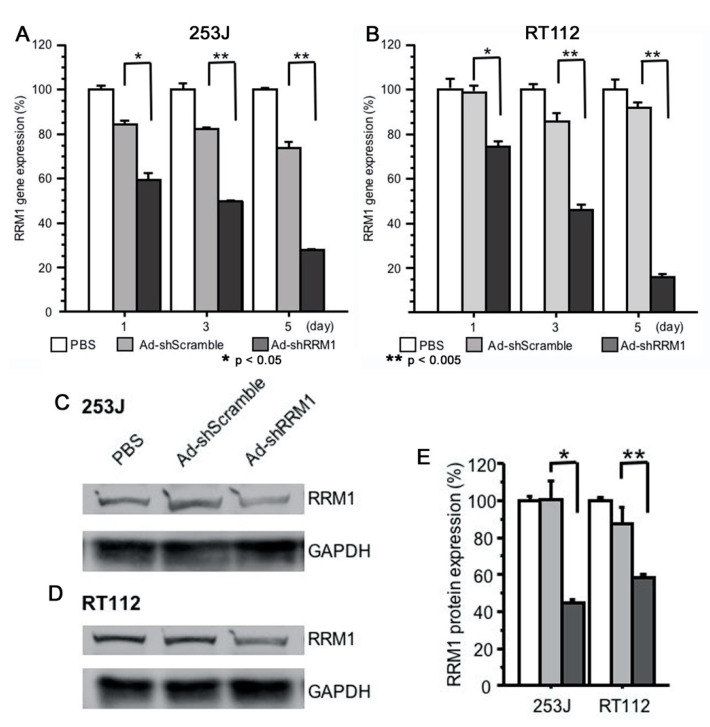
RRM1 gene and protein expression in RRM1 highly expressed human bladder cancer cells after transfection with adenoviral vectors. RRM1 gene expressions were effectively inhibited after Ad-shRRM1 infection at MOI = 10 in (**A**) 253J cells and (**B**) TR112 cells, time-dependently. RRM1 protein expression can be seen in RRM1 highly expressed (**C**) 253J and (**D**) RT112 cells. Quantification of the PBS, Ad-shScramble and Ad-shRRM1 ration of RRM1/GAPDH protein expression (**E**). Western blot analyses were performed at 72 h after transduction with adenoviral vectors at MOI = 20 and one of three experiments with similar results is shown. (MOI: multiplicity of infection; * *p* < 0.05, ** *p* < 0.005 vs. Ad-shScramble treatment).

**Figure 4 ijms-22-04102-f004:**
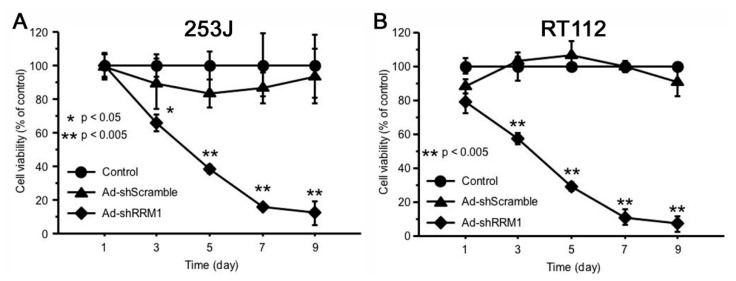
Ad-shRRM1 significantly inhibits cell viability of RRM1-expressing bladder cell lines. Cell viability was evaluated by the 3-(4,5-dimethylthiazol-2-yl)-2,5-diphenyltetrazolium bromide (MTT) assay after transduction with adenoviral vectors at MOI = 10 at series time point of 1, 3, 5, 7 and 9 day. (**A**) results in 253J cells; (**B**) results in RT112 cells. (MOI: multiplicity of infection; * *p* < 0.05, ** *p* < 0.005 vs. Ad-shScramble treatment).

**Figure 5 ijms-22-04102-f005:**
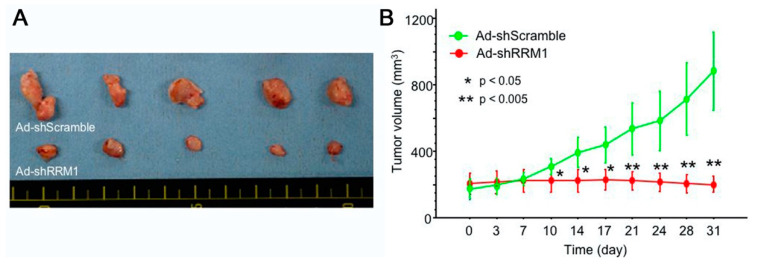
In vivo anti-tumor effect of Ad-shRRM1 on RRM1 highly expressed RT112 xenografts. RT112 tumors, treated with Ad-shScramble or Ad-shRRM1, were enucleated 31 days after initiation of treatment. Images show (**A**) the appearance and (**B**) tumor volumes of RRM1 highly expressed RR112 xenografts in nude mice. (*: *p* < 0.05 versus Ad-shScramble treatment group).

**Figure 6 ijms-22-04102-f006:**
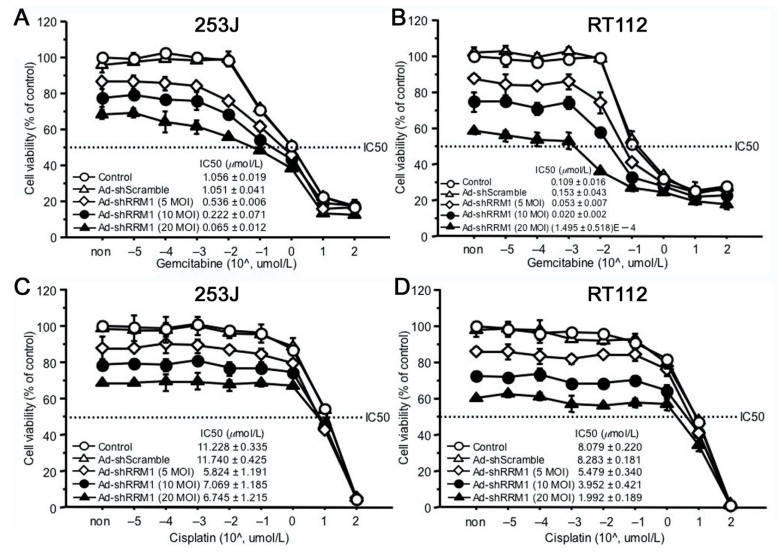
The anti-cancer effects of combination of Ad-shRRM1 and GEM on RRM1 highly expressing bladder cancer cells. RRM1 inhibition with Ad-shRRM1 specifically sensitizes RRM1 highly expressed 253J (**A**) and RT112 cells (**B**) to GEM, but not to CDDP (**C**) and (**D**), respectively. Cells were infected with Ad-shRRM1 at series MOI of 5, 10, and 20 for 24 h, then treated with GEM or CDDP at gradually increasing concentrations. IC_50_ was evaluated by MTT assay 3 days after drug administration. Cell viability at starting point of non indicates the pure effect of adenoviral vector on the cell proliferation reference to cell viability treated with medium respectively. (IC_50_: drug concentration that inhibited cell growth by 50%; MOI, multiplicity of infection; GEM: gemcitabine; CDDP: cisplatin).

## Data Availability

Not applicable.

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
