# Peer review of "Knockdown of RRM1 with Adenoviral shRNA Vectors to Inhibit Tumor Cell Viability and Increase Chemotherapeutic Sensitivity to Gemcitabine in Bladder Cancer Cells"

_ijms, 2021, doi:10.3390/ijms22084102_

Round 1

Reviewer 1 Report

The authors have submitted a MS to demonstrate the RMM1 siiencing inhibits bladder cancer cells viability and can increase cell sensitivity to Gemcitabine.

The MS is clear, the results are convincing and the discussion consistent

The link between resistance to Gem sensitivity and RMM1 is not a novelty.

In the context of bladder cancer,  the Authors should discuss the potential interaction between RMMI and other gene involved into DNA repair could be hypothesised

Author Response

Thank you very much for your careful review and nice comments on our manuscript. We also think the potential interaction between RMM1 and other genes involved into DNA repair should be very important for understanding the mechanism on the role of RRM1 and GEM resistance. We are planning to explore the relation between RRM1 and genes such as p53 and RARP in the near future.

In addition, we have added the discussion regarding the potential interaction between RRM1 and other factors influencing GEM resistance in the revised manuscript (Line 201-204).

Reviewer 2 Report

The authors presented a manuscript “Knockdown of RRM1 with Adenoviral shRNA Vectors to Inhibit Tumor Cell Viability and Increase Chemotherapeutic Sensitivity to Gemcitabine, in Bladder Cancer Cells”.

The rationale of this work is to verify the efficacy of RRM1 gene silencing in sensitizing bladder cancer cells to gemcitabine. This study has been performed using experimental approaches in vitro, using two bladder cancer cell lines, and in vivo using xenograft implants in nude mice. The results here reported show that the silencing of the RRM1 gene alone was able to reduce the viability of bladder cancer cells with high RRM1 expression, making them more susceptible to the action of Gemcitabine treatment. In vivo, injection of ad-shRRM1 recombinant adenovirus blocked the growth of the implanted tumor in nude mice.

The experimental design is appropriate, the results are convincing, and support similar published data obtained from other cancer cell lines. The work is original in the application of RRM1 knockdown in reducing the tumorigenic potential of bladder cancer cells, although it is not in an absolute sense in the field of cancer study.

I have only minor concerns to bring to the attention of the authors:

-The text contains several grammatical errors and I invite the authors to review the manuscript by a native speaker

Abstract

-Line 17, two high RRM1-overexpressing cell lines—253J and RT112—were…

-Line 18, An RRM1-targeting shRNA cDNA was cloned into an adenoviral vector…

Results

Line 71, please indicate the control used as 100% to which samples with 28.6% to 153.5% have been compared; Indicate which cell line correspond 28.6 % and 153.5%.

-Figure 3A, the control is reported with a value greater than 100%.

-Line 101, please delete “or AD-shScramble”

- Line 143, Figure 6, in my opinion, there is a discrepancy between the cell viability values in the untreated samples (non) transduced with Ad-shRRM1 in the two graphs (upper and lower panels) corresponding to the same cell line. How do the authors explain this discrepancy? Please compare these results with the values reported in Figure 3, at the same period of incubation (Day 3).  Moreover, it is difficult to understand how these values have been calculated and reported in the graph. For a better understanding of the graph, each set of values (Control, Ad-ShScramble, Ad-ShRRM1) should be reported as percentage of the corresponding untreated cells (non in Figure), so all curves start at 100%. This must be clearly stated in the legend of the figure. The values of cell viability in Ad-ShScramble and Ad-ShRRM1 cells (as a percentage with respect to the Control at day 3) should be indicated both in the text and in the legend of figure 4.

Material and methods:

-Line 235, 238, Please correct “GPR87”

-Line 240, Please correct U nucleotides in T.

-Lines 241-242, Please insert a reference where a detailed description of recombinant adenovirus particles for transduction used in this work is reported.

-Line 260, please correct “Two cells…” with “Two cell lines”

-Line 278, please insert the reference to the approval of experimental protocol for in vivo study in animals by the local ethics committee.

Discussion:

- Why was gemcitabine treatment not performed in Sh-adRRM1 mice?

Author Response

Comments and Suggestions for Authors

The authors presented a manuscript “Knockdown of RRM1 with Adenoviral shRNA Vectors to Inhibit Tumor Cell Viability and Increase Chemotherapeutic Sensitivity to Gemcitabine, in Bladder Cancer Cells”.

The rationale of this work is to verify the efficacy of RRM1 gene silencing in sensitizing bladder cancer cells to gemcitabine. This study has been performed using experimental approaches in vitro, using two bladder cancer cell lines, and in vivo using xenograft implants in nude mice. The results here reported show that the silencing of the RRM1 gene alone was able to reduce the viability of bladder cancer cells with high RRM1 expression, making them more susceptible to the action of Gemcitabine treatment. In vivo, injection of ad-shRRM1 recombinant adenovirus blocked the growth of the implanted tumor in nude mice.

The experimental design is appropriate, the results are convincing, and support similar published data obtained from other cancer cell lines. The work is original in the application of RRM1 knockdown in reducing the tumorigenic potential of bladder cancer cells, although it is not in an absolute sense in the field of cancer study.

I have only minor concerns to bring to the attention of the authors:

-The text contains several grammatical errors and I invite the authors to review the manuscript by a native speaker

: Thank you very much for your careful review and commends on our manuscript. We revised our manuscript carefully according to you commend and reviewed the revised manuscript by a native speaker.

Abstract

-Line 17, two high RRM1-overexpressing cell lines—253J and RT112—were…

: These “—” have been omitted in the revised manuscript (Line 17).

-Line 18, An RRM1-targeting shRNA cDNA was cloned into an adenoviral vector…

: This sentence has been modified in the revised manuscript (Line 18).

Results

Line 71, please indicate the control used as 100% to which samples with 28.6% to 153.5% have been compared; Indicate which cell line correspond 28.6 % and 153.5%.

: We have added this information in the revised manuscript (Line 74, Line 82 to Line 84).

-Figure 3A, the control is reported with a value greater than 100%.

: We are sorry for this mistake during preparing this Figure. We have corrected the mistake in the revised manuscript (Figure 3A).

-Line 101, please delete “or AD-shScramble”

: We have delated the phrase “or AD-shScramble” in the revised manuscript (Line 107).

- Line 143, Figure 6, in my opinion, there is a discrepancy between the cell viability values in the untreated samples (non) transduced with Ad-shRRM1 in the two graphs (upper and lower panels) corresponding to the same cell line. How do the authors explain this discrepancy? Please compare these results with the values reported in Figure 3, at the same period of incubation (Day 3).  Moreover, it is difficult to understand how these values have been calculated and reported in the graph. For a better understanding of the graph, each set of values (Control, Ad-ShScramble, Ad-ShRRM1) should be reported as percentage of the corresponding untreated cells (non in Figure), so all curves start at 100%. This must be clearly stated in the legend of the figure. The values of cell viability in Ad-ShScramble and Ad-ShRRM1 cells (as a percentage with respect to the Control at day 3) should be indicated both in the text and in the legend of figure 4.

: Thank you very much for your revise on our manuscript.

The discrepancy between the cell viability values in the untreated samples (non) transduced with Ad-shRRM1 in the two graphs (upper and lower panels) is caused by the different time period for these tow experiments for GEM and CDDP. We have managed to repeat these experiments and hope to supply more clear figures soon. For the combination experiment of Ad-shRRM1 and GEM, the time period of incubation is acutely 4 days but not 3 day as the experiment showed in Figure 4. In the experiments presented in Figure 6, cells were firstly infected with adenoviral vector or medium (non) 1 day before drug administration, then treated with GEM, CDDP or medium (non) for 3 days before MTT evaluation. In addition, as mentioned in the text, due to the strong inhibitory effect of Ad-shRRM1 on cell proliferation, we could not obtain the IC50 of GEM after Ad-shRRM1 infection at the same concentration (10 MOI) as Figure 4. We have reduced the Ad-shRRM1 exposure by half, to MOI of 5. Those factors may be the reason for the discrepancy of cell viability between Figure 6 and Figure 4. We have modified these descriptions in the revised manuscript (Line 123-124 and Line 156-157).

Material and methods:

-Line 235, 238, Please correct “GPR87”

: These mistakes have been corrected in the revised manuscript (Line 248, 251).

-Line 240, Please correct U nucleotides in T.

: We have corrected U nucleotides in T in the revised manuscript (Line 249-251).

-Lines 241-242, Please insert a reference where a detailed description of recombinant adenovirus particles for transduction used in this work is reported.

: We have inserted references in the revised manuscript (Lines 256).

-Line 260, please correct “Two cells…” with “Two cell lines”

: We have corrected “Two cells…” with “Two cell lines” in the revised manuscript (Lines 274).

-Line 278, please insert the reference to the approval of experimental protocol for in vivo study in animals by the local ethics committee.

: We have inserted references to the approval of experimental protocol for in vivo study in animals by the local ethics committee in the revised manuscript (Line 293).

Discussion:

- Why was gemcitabine treatment not performed in Sh-adRRM1 mice?

: In fact, in the current experiment, Ad-shRRM1 treatment alone had already potently inhibited tumour growth, and the effect would have been difficult to evaluate if GEM had been added, so no experiments with GEM were performed. If in vivo experiments are necessary, we hope to treat ad-shRRM1-treated xenograft mouse models and controls with GEM in the near future to clarify the interaction between Ad-shRRM1 and GEM.

Reviewer 3 Report

Comments:

In this manuscript, Zhang et al. elucidate the application of adeno viral vector expressing shRNA against bladder cancer cells that exhibit high expression of RRM1. They perform qPCR, western blotting and proliferation assays-based analyses coupled with a xenograft experiment to support their claim. The authors further claim a synergistic effect of GEM with Ad-shRRM1 in the bladder cancer cells and propose this as a potential option for therapeutic intervention against bladder cancer. Overall, this manuscript is a well-done study, however there are some potential concerns that the authors should address. I recommend that this be considered for publication after the authors address the following comments.

Major comments:

  1. In Figure1, how does the expression of RRM1 in bladder cancer cells compare to that of normal urothelial cells? An mRNA or protein expression level analysis of this sort can determine how specific RRM1 is for bladder cancer cells.
  2. What happens to normal urothelial cells when RRM1 is silenced? In Figure 3, a cytotoxicity control is strongly recommended i.e., using normal urothelial cells as control compared to the bladder cancer cells.
  3. In Figure 3C and D, it is highly recommended to show WB at MOI of 10 and the time points used for Fig 1A and B i.e., 1, 3 and 5 days to assess the knockdown efficiency at these points. Ideally, WB with quantification at these three time points can make the claims stronger.
  4. Figure 6A and B are not very convincing. The authors need to explain why the starting points of assessment in the plots (“non” point), Ad-shRRM1 start at a very low point compared to Control and Ad-shScramble. I would assume all three starting data points are supposed to start at the same point or within the error range to truly accept the sensitivity fold changes. This needs to be addressed.
  5. In Figure 6, the authors use a higher MOI to obtain a modest sensitivity raise, further raising concerns about specificity and cytotoxicity.
  6. The data in Figure 6 with claims about synergy are not convincing. For making a claim about synergy, I strongly suggest the authors perform a xenograft experiment with GEM, Ad-shRRM1 and GEM + Ad-shRRM1. This will bolster the claims so this needs to be addressed.  
  7. There is insufficient data shown to support a claim of drug combination synergy. For instance, Meyer et al. describe a more contemporary approach to quantify drug combination synergy (DOI: 10.1016/j.cels.2019.01.003). Therefore, clarify if the effects the authors observe is synergistic or additive.

Minor comments:

  1. The language of line 49-50 is unclear. “This deoxy…..activity” should be made clear.
  2. Have people performed GEM treatment and analysis on healthy urothelial cells? Recommended to include that in the discussion.
  3. In general, the term “overexpression/overexpressing” implies usage of exogenous DNA to overexpress a gene of interest in the cells via transfection or transduction. Therefore, I suggest using “highly expressed” or something similar in lieu of “overexpressing” so that it is not confusing. I urge the authors to clarify this language everywhere appropriate.
  4. Lines 181-182 of discussion suggests a very broad claim which is not fully supported with the current set of data. I urge the authors to reconsider this after the recommended experiments.

Author Response

Comments:

In this manuscript, Zhang et al. elucidate the application of adeno viral vector expressing shRNA against bladder cancer cells that exhibit high expression of RRM1. They perform qPCR, western blotting and proliferation assays-based analyses coupled with a xenograft experiment to support their claim. The authors further claim a synergistic effect of GEM with Ad-shRRM1 in the bladder cancer cells and propose this as a potential option for therapeutic intervention against bladder cancer. Overall, this manuscript is a well-done study, however there are some potential concerns that the authors should address. I recommend that this be considered for publication after the authors address the following comments.

Major comments:

  1. In Figure1, how does the expression of RRM1 in bladder cancer cells compare to that of normal urothelial cells? An mRNA or protein expression level analysis of this sort can determine how specific RRM1 is for bladder cancer cells.

: Thank you very much for your comment. We are regret for this absence of normal urothelial cell. We have already managed to purchase normal urothelial cell line and hope to supply this information soon.

  1. What happens to normal urothelial cells when RRM1 is silenced? In Figure 3, a cytotoxicity control is strongly recommended i.e., using normal urothelial cells as control compared to the bladder cancer cells.

: As mentioned in above, we have already managed to purchase normal urothelial cell line and hope to supply this information soon.

  1. In Figure 3C and D, it is highly recommended to show WB at MOI of 10 and the time points used for Fig 1A and B i.e., 1, 3 and 5 days to assess the knockdown efficiency at these points. Ideally, WB with quantification at these three time points can make the claims stronger.

: Thank you very much for your suggestion. We have raised frozen cells for assessing the knockdown efficiency at these points and hope to supply this information in time. As we know, there is a discrepancy in expression between gene expression and protein expression. I have planned to carry out an additional experiment. However, the slight difference in gene expression of 10% between the 1st and 3rd days of 253J cell may be difficult to be judged on the WB.

  1. Figure 6A and B are not very convincing. The authors need to explain why the starting points of assessment in the plots (“non” point), Ad-shRRM1 start at a very low point compared to Control and Ad-shScramble. I would assume all three starting data points are supposed to start at the same point or within the error range to truly accept the sensitivity fold changes. This needs to be addressed.

: We tried to use the “non” point as the starting point in the plot for suppling information regarding the pure inhibitory effect of adenoviral vector on the cell proliferation. For the combination experiment of Ad-shRRM1 and GEM, cells were firstly infected with adenoviral vector or medium (non) 1 day before drug administration, then treated with GEM, CDDP or medium (non) for 3 days before MTT evaluation. In addition, as mentioned in the text, due to the strong inhibitory effect of Ad-shRRM1 on cell proliferation, we could not obtain the IC50 of GEM after Ad-shRRM1 infection at the same concentration (10 MOI) as Figure 4. We have reduced the Ad-shRRM1 exposure by half, to MOI of 5. The discrepancy between the cell viability values in the untreated samples (non) transduced with Ad-shRRM1 in the two graphs (upper and lower panels) is caused by the different time period for these tow experiments for GEM and CDDP. Those factors may be the reason for the discrepancy of cell viability in the Figure 6 and Figure 4. We have managed to repeat these experiments and hope to supply more clear figures soon. We have modified these descriptions in the revised manuscript (Line 123-124 and Line 156-157).

  1. In Figure 6, the authors use a higher MOI to obtain a modest sensitivity raise, further raising concerns about specificity and cytotoxicity.

: The concentration of Ad-shRRM1 in the Figure 6 should be 5 but not 20 MOI. We are so sorry for this mistake. As mentioned in the text, due to the strong inhibitory effect of Ad-shRRM1 on cell proliferation, we could not obtain the IC50 of GEM after Ad-shRRM1 infection at the standard MOI of 10, in these cells (Line 180-181). We reduced the Ad-shRRM1 exposure, to 5  Mol. We have corrected these mistakes in the revised manuscript (Line182) and Figure 6 (Line 154).

  1. The data in Figure 6 with claims about synergy are not convincing. For making a claim about synergy, I strongly suggest the authors perform a xenograft experiment with GEM, Ad-shRRM1 and GEM + Ad-shRRM1. This will bolster the claims so this needs to be addressed.

: Thank you very much for your suggestion. Unfortunately, we have not data for the combination of Ad-shRRM1 with Gemcitabine in the mice this time. As your suggestion, the in vivo experiment with GEM should be very important. We are planning to complete these important data in the near future.

  1. There is insufficient data shown to support a claim of drug combination synergy. For instance, Meyer et al. describe a more contemporary approach to quantify drug combination synergy (DOI: 10.1016/j.cels.2019.01.003). Therefore, clarify if the effects the authors observe is synergistic or additive.

: Actually, it is not sufficient for us to just the drug combination relation as synergy with the present data. We have modified the description in the revised manuscript (Line 151-152). Thank you very much for the paper by Meyer et al. you suggested, we will try to clarify the correlation between Ad-shRRM1 and GEM with this new method.

Minor comments:

  1. The language of line 49-50 is unclear. “This deoxy…..activity” should be made clear.

: We have modified this description in the revised manuscript (Line 49-53).

  1. Have people performed GEM treatment and analysis on healthy urothelial cells? Recommended to include that in the discussion.

: Thank you for your suggestion. We do not have the in vitro and in vivo data of normal urothelial cells treated with gemcitabine. However, the intravesical instillation of gemcitabine has been tested in two animal models and in two phase 1 trials. In all four studies it was concluded that local and systemic side effects were minimal. We add this information in this manuscript (Line193-196).

  1. In general, the term “overexpression/overexpressing” implies usage of exogenous DNA to overexpress a gene of interest in the cells via transfection or transduction. Therefore, I suggest using “highly expressed” or something similar in lieu of “overexpressing” so that it is not confusing. I urge the authors to clarify this language everywhere appropriate.

: We have corrected the term “overexpression/overexpressing” with “RRM1 high expressed” in the revised manuscript accordingly.

  1. Lines 181-182 of discussion suggests a very broad claim which is not fully supported with the current set of data. I urge the authors to reconsider this after the recommended experiments.

: We have restricted our claim in the revised manuscript (Line 187).

 [HI1]

Round 2

Reviewer 3 Report

While the manuscript seems to have been revised as suggested, some key concerns still linger.

At the current state, Figure 6 still doesn't look very convincing with only legend changed but the plots remain the same. Given that the in vivo drug+shRNA combination experiment is not performed, the authors are strongly advised to carry out the suggested in vitro experiments for the first submission. Those experiments seem to be achievable and will help bolster the claims. Once those experiments are reported, the manuscript should be strongly considered for publication.

Author Response

Thank you very much for your kind suggestion. We have carefully studied your comments and we agree that the MuSyC framework is a very good way to distinguish the synergistic effects of two types of drugs, but it is very difficult for us to complete such experiments, because our university does not have such equipment. To improve our data as much as possible, we have repeated all of the combination experiments, added two Ad-RMM1 groups of low concentrations and tried to bolster the claims with a series Ad-RRM1 expose in these two cell lines. On the first day, cells were first infected with different concentrations of Ad-shRRM1, to knock down the RRM1 gene. After of 24hr, GEM or CDDP treatment was then applied. On the fourth day, the IC50 were determined by MTT array. The 253J cell viability was 86.9 ± 1.2%, 77.5 ± 3.5% and 68.2 ± 1.6% infected with Ad-shRRM1 at 5, 10 and 20 MOI 4 day after infection (Figure 6A). The RT112 cell viability was 87.7 ± 1.5%, 74.6 ± 4.1% and 58.0 ± 0.7% infected with Ad-shRRM1 at 5, 10 and 20 MOI 4 day after infection (Figure 6B). Double increase in Ad-shRRM1 exposure resulted about 9% and 12% cell inhibition for 253J and RT112 cell respectively. On the other hand, when in combination with GEM treatment, in case of Ad-shRRM1 exposure at MOI of 20, the sensitivity to GEM (IC50) increased significantly and was 27.5 times higher in 253J cell and 7,712 times higher in RT112 cell (Figure 6A and 6B). The Figure 6 have been revised and discussed in the revised manuscript (Line 180-186). We consider the combined relationship between Ad-RRM1 and GEM as a synergistic enhancement effect. We also hope to further demonstrate our results in the form of MuSyC framework in the near future as you suggested.

Round 3

Reviewer 3 Report

The authors have clarified the previously raised concerns. The addition of more data points with higher MOI in Fig 6 bolster their claims.